# Determination of Various Drying Methods’ Impact on Odour Quality of True Lavender (*Lavandula angustifolia* Mill.) Flowers

**DOI:** 10.3390/molecules24162900

**Published:** 2019-08-09

**Authors:** Jacek Łyczko, Klaudiusz Jałoszyński, Mariusz Surma, José Miguel García-Garví, Ángel A. Carbonell-Barrachina, Antoni Szumny

**Affiliations:** 1Faculty of Biotechnology and Food Science, Wrocław University of Environmental and Life Sciences, 50-375 Wrocław, Poland; 2Institute of Agricultural Engineering, Wrocław University of Environmental and Life Sciences, 51-630 Wrocław, Poland; 3Department Agro-food Technology, Escuela Politécnica Superior de Orihuela, Universidad Miguel Hernández de Elche, Elche, 03312 Alicante, Spain

**Keywords:** essential oils, drying methods, olfactometry, terpenoids, true lavender flowers, quality, volatile profile, SPME

## Abstract

True lavender flowers (*Lavandula angustifolia* Mill.) is a critical source of essential oils and a flavouring agent used in numerous industries like foods, cosmetics and pharmaceuticals. Its main volatile constituents are linalool and linalyl acetate, which are commonly considered as main odour-active constituents (OACs). Nevertheless, the quality of true lavender flowers is highly dependent on its post-harvest treatment, mainly the preservation method. Recognising that drying is the most frequently used preservation method, the influence of various drying methods, including convective drying (CD) at 50, 60 and 70 °C, vacuum-microwave drying (VMD) with powers 240, 360 and 480 W and combined convective pre-drying at 60 °C followed by vacuum-microwave finish-drying with power 480 W (CPD-VMFD), on the quality of true lavender flowers was verified. The evaluation of influence was carried out by HS-SPME(HS, solid-phase microextraction), GC-MS, GC-MS-O (gas chromatography–mass spectrometry–olfactometry) techniques. Moreover, the sensory panel has assessed the sample odour quality. As a result, the optimal drying methods regarding the requirements for products were established. Overall, for total essential oil recovery, CD at 50 °C is the optimal drying method, while for odour quality concerning the sensory panel evaluation, VMD with power 360 W combined CPD-VMFD and CD at 50 °C is the optimal drying method.

## 1. Introduction

*Lavandula* genus is a large group of tremendously useful plants and, along with such herbs like rosemary, basil or sage, belongs to the *Lamiacae* family. The most well-known representative of *Lavandula* sp. is *Lavandula angustifolia* Mill.—true lavender. Its visibility is caused by numerous applications, both of the plant and its derivatives in pharmacy, aromatherapy, perfumes and cosmetics, food flavouring and preservation, household products or just as a decorative plant [1,2]. True lavender plants, with its characteristic violet flowers and narrow leaves, are native to Europe, mainly the Mediterranean area, North America and Australia. Nowadays, the largest cultivations are located in France, Bulgaria and Turkey. In Bulgaria its cultivation area reaches more than 6000 ha, and the Turkish production of lavender flowers oscillates around 50–75 tons per ha [3,4,5,6].

The history of true lavender usage goes back to Greek and Roman civilizations in ancient times, which had used it due to its flavour. In the Middle Ages, lavender and its essential oil (EO) were recognized as a potential drug. It was applied in cases of migraines, panic attacks or heart problems [5]. More contemporary applications are related to the neuroprotective and anti-aging [7,8], preservative (due to antioxidant and antibacterial) [9,10], sedative [11] and anti-insomnia [12] properties of true lavender EO. However, the most important feature of lavender is its characteristic odour, which is exploited widely in the cosmetics and food industries. In the case of cosmetics, the recent research scopes are focused on applying lavender free and encapsulated EO or hydrosols for aromatic, antioxidant and antimicrobial purposes [9,13,14,15]. True lavender and its derivatives have found similar functions in the food industry. The most effects are expected in the case of food microbial control, for instance *Botritis cinereal*, *Escherichia coli*, *Staphylococcus aureus*, *Pseudomonas auerginosa* or *Candida albicans* [16,17], preservation fatty acids before oxidation [18] and flavouring nanocapsules [19].

These wide applications are mainly related to unique lavender flowers EO composition. Numerous sources show that its main volatile constituents are linalool (20.0%–45.0%), linalyl acetate (25.0%–46.0%) and lavandulyl acetate (>0.2%) [2,5,20]. Nevertheless, other components that occur in lavender EO such as camphor, 1,8-cineole, borneol, ocimene and in higher amounts may be a disqualifying factor for expected plant usage [21]. 

The quality and effectiveness of true lavender flowers or their derivatives are remarkably related to various factors like plant chemotype, conditions of harvest and cultivation and post-harvest treatment like storage and preservation method [22]. For economic efficiency, the most popular method of obtaining high-quality products is to use the hot air–convective drying (CD). Nevertheless, because one wants to reduce the loss of fragile compounds, other methods like vacuum-microwave drying (VMD), freeze-drying (FD), infrared drying (IRD), spray drying (SpD) or some combinations like convective pre-drying with vacuum-microwave finish drying (CPD-VMFD) may be considered [23,24].

Since in previous work by Łyczko et al. (2019) [25] the influence of various drying methods on the quality of true lavender leaves was examined, it was decided to continue the investigation on true lavender flowers from the same cultivar harvest in Poland in 2018. It is an important approach given the significance of true lavender flowers’ quality, which is dependent on total EO content and the main constituents’ contribution in total EO and in volatile profile, and regarding the raw material structural difference and behaviour during the drying process. Considering the post-harvest process cost efficiency and for convenience, the process the same drying methods were applied to enable the same treatment for leaves and flowers. However, more analytical techniques were used for true lavender flowers, like sensory analysis in comparison olphactometry. In this case, this study aimed to evaluate various drying techniques and/or their specific parameters by examining their effect on true lavender flower EO, headspace (HS) volatile composition, EO composition and sensory quality. The drying methods evaluated included CD, VMD and CPD-VMFD, and different conditions were assayed (temperature, time and wattage). To extract the plant material hydrodistillation with Deryng apparatus and HS, solid-phase microextraction (HS-SPME) were applied. Then, the volatile composition analyses were performed by the gas chromatography-mass spectrometry (GC-MS) technique. In addition, the sensory analysis of dried plant material was carried out within the evaluation of the variability of odour-active constituents (OACs) present in true lavender flowers.

## 2. Results and Discussion

### 2.1. Drying Kinetics

Figure 1 shows changes with time of the moisture ratio (MR) of sample flowers dehydrated by VMD at three magnetron powers (240, 360 and 480 W, Figure 1a), CD at temperatures in the range 50 to 70°C (Figure 1b), CPD-VMFD consisting of CD at 60°C and VMD at magnetron powers of 480 W (Figure 1c). The drying times, together with the maximum temperatures, the final moisture content and the constants of the Page model are shown in Table 1.

The Page model can be successfully used to describe the drying kinetics of the true lavender flower dehydrated by the CD, VMD and CPD-VMD methods, characterized by high values of the determination coefficient (R^2^ > 0.99) and low root-mean squared error (RMSE) values (<0.05). A good adaptation of the applied Page model to the description of the drying kinetics was found in many earlier publications of dill leaves, true lavender leaves, chanterelle and oyster mushrooms [25,26,27,28]. 

In the case of CD, increasing the drying air temperature from 50 to 70 °C decreased the time of drying from 245 to 150 minutes, respectively. With respect to VMD, radical reductions in the total drying time have been observed: the time was shortened from 44 to 20 minutes with a power change from 240 to 480 W. This radical reduction in the total drying time of VMD compared to CD is a result of the conventional water diffusion occurring according to Fick’s law, which is supported by a pressure diffusion mechanism of the Darcy type [29]. Combined CPD and VMFD using 480 W shortened the drying time of flowers almost four-fold compared to CD at 50°C. The use of CD and 480 W power caused a drop in the material temperature during VMD by 4 °C for flowers in reference to VMD 480 W. This condition is caused by the molecular distribution of water particles inside the dried material, and the distribution of water particles has an effect on the generation of heat energy under microwave radiation during VMD [30,31]. Energy consumption during the CD of plant materials is much lower than in VMD [32,33]. In industrial conditions, the best solution is combined drying consisting of CPD and VMFD. The CD is very effective at the beginning of the drying process (the largest loss of water occurs during the phase) and VMD at the final stage of drying (removal of water strongly bound to the cellular structure of the material being dried) [23,30,31]. The final choice of recommendations drying process should be related to the aspects of the dried material (volatile composition and sensory attributes) [30].

### 2.2. Volatile Profile of True Lavender Flowers Cultivated in Poland

The volatile profile of true lavender flowers consisted of seventy-four constituents, among which seventy-two were identified (the mass spectra of unidentified compounds are available in Appendix A). This high number of volatile compounds agreed quite well with results obtained by Śmigielski et al. (2018) [34], which had identified sixty-seven compounds in the true lavender flower EO. However, they achieved higher sensitivity in the case of dried flowers, which may be caused by applying different analytical methods. It should be underlined that they had used true lavender flowers harvest in 2013, delivered by the same supplier that was used in this study (harvest in 2018). 

Other major constituents revealed in this study were *cis*-β-ocimene (4.97% ± 0.74%), *trans*-β-ocimene (3.99% ± 0.11%), 1-octen-3-ol acetate (1.57% ± 0.20%), terpinen-4-ol (3.36% ± 0.29%), lavandulyl acetate (5.17% ± 0.48%), caryophyllene (7.57% ± 1.45%) and alloaromadendrene (3.80% ± 0.76%). All of these constituents are reported in ISO and pharmacopeial standards for true lavender EO [1]. All results are listed in Table 2.

Other studies that applied HS-SPME analysis with DVB/CAR/PDMS (Divinylbenzene/Carboxen/Polydimethylsiloxane) fibre had revealed more volatile constituents profile similar to one obtained in this study. Da Porto and Decorti (2008) [36] found higher amounts of linalyl acetate over linalool in true lavender flowers collected in Middle-Friuli (Italy), while Fu et al. (2018) [37] obtained results from some true lavender varieties (French blue, H-701 and C-179(2)) collected in China, similarly to this study. Moreover, some other constituents in less amounts like *cis*-β-ocimene, *trans*-β-ocimene, lavandulyl acetate [37] and caryophyllene [36] also overlap in this and mentioned studies. Nevertheless, Da Porto and Decorti (2008) [36] also report high amounts of eucalyptol (7.95%–10.78%) and camphor (8.60%–12.59%) found in the volatile profile of true lavender flowers, which is in opposition for this and to our recent study regarding the volatile profile of true lavender leaves [25]. It appears that, for the chemotype of true lavender used in these studies, the characteristic is high ratio camphor/eucalyptol and low amounts of linalool and linalyl acetate in the volatile profile of true lavender leaves, and the opposite situation in the volatile profile of true lavender flowers. Results reported by Marin et al. (2016) and Hajhashemi et al. (2003) supported that true lavender leaves are abundant in camphor and eucalyptol, while true lavender flowers are richer in linalool and linalyl acetate. [38,39]

### 2.3. Influence of Various Drying Techniques on the Quality of True Lavender Flowers Cultivated in Poland

EO content of true lavender flowers obtained with Deryng apparatus was 5.18%, which is consistent with the findings of Dušková et al. (2016) [40], who obtained on average of 6.52% of EO cultivated in the Czech Republic. Moreover, also according to Aprotosoaie et al.*’s* (2017) [1] overview, this result is within the boundaries reported by numerous studies. It should be underlined that steam distillation, rather than hydrodistillation, is applied in industry practice. Despite that, for analytical purposes in laboratory scale, it is common to apply hydrodistillation with Clevenger or Deryng apparatus, which is supported by methods used in numerous works [41,42,43,44,45,46,47]. However, there may be an observed difference in the distribution of major constituents. In the current study, similar amounts of linalyl acetate (18.98% ± 0.33%) and linalool (18.37% ± 0.12%) were found, which is in opposition to Śmigielski et al.*’s* (2018) [34] result (respectively 19.7%–23.4% and 26.5%–34.7%). This result may again be caused by different years of cultivation of true lavender flowers. Nevertheless, applied drying methods had significantly affected the EO composition and major volatile constituents’ retention, which is presented in Table 3 (list of all identified constituents of true lavender flowers EO is available in Appendix A). The most efficient drying methods, which are not significantly different, were CD at 50 °C and 70 °C or CPD-VMFD, although those methods still decreased the EO yield from 3.8 to 4.5 times. This result is higher than the one reported by Prusinowska and Śmigielski (2015) [48], however, they considered a different drying method—shade drying. Also, Sadowska (2012) [49] had pointed out that naturally dried lavender was characterized by a higher amount of EO than that dried mechanically. On the contrary, Figiel et al. (2010) [50] and Nöfer et al. (2018) used the same drying methods respectively on oregano (*Origanum vulgare*) and king bolete (*Boletus edulis*) and obtained results similar to those found in this study. In the case of major volatile constituents of EO, the most efficient drying method (not for all compounds) overlapped with the most efficient method for total EO retention, although for most CD at 50 °C it was still the most attractive one. This result is in contrast to our previous work, focused on true lavender leaves [25], where the CD at 50 °C was the less effective for total EO. Nevertheless, it is worth underlining that, in the case of true lavender flowers, twelve out of fourteen compounds demonstrate the highest retention for this drying method. These different results may be caused by differences in the morphological structure of true lavender leaves and flowers. Also Chua et al. (2019a) [42] and Chua et al. (2019b) [51] had observed more efficient results for CD than for VMD drying methods.

The drying process had also affected the volatile constituents present in the volatile profile of true lavender flowers determined by HS-SPME technique, which is presented in Table 4. Obtained results differ from results for EO constituents. The efficient drying method is more diversified for specific constituents. In the case of two major odour-shaping constituents of true lavender flowers, linalool and linalyl acetate, respectively CD at 50 °C and CD at 60 and 70 °C, improved the highest compounds share in volatile profile, which partially coincides with EO analysis results. Overall, these results are similar to those obtained for true lavender leaves [25].

### 2.4. Odour-Active Compound Determination and Sensory Value of True Lavender Flowers Affected by Various Drying Methods

GC-MS-Olfactometry analysis of true lavender flower EO had revealed that only ten constituents from seventy-four identified in its volatile profile may be classified as OACs and are presented in Table 5 within their aroma description and variability. The majority of them were also identified by Xiao et al. (2017) [52] although they identified thirty constituents as OACs in total. The difference may be caused by the origin of the true lavender plants, among other factors. Moreover, in this study OACs were identified for dried true lavender flowers.

The sensory panel results, given in Figure 2, had shown that true lavender flowers dried by the VMD 360 W method were recognized as the material with the most intense odour, while the CD 70 °C method was the less suitable since it had the lowest intensity (.pdf file with all ranks is available in Appendix A). Also, with lower ranks but not scientifically different was the assessment of combined CPD-VMFD and CD at 50 °C products. These results were surprising because they do not agree with the total volatile contents discussed previously here, where CD methods were more effective than VMD methods in retaining a higher content of EO. Such findings demonstrated that the highest amounts of constituents or their share in a volatile profile do not always guarantee the best sensory quality. It seemed that, in the case of major true lavender flower OACs, the most favourable linalool:linalyl acetate ratio is approximately 1:2, while linalyl acetate share is around 43%–47%. Similar findings were reported by Beale et al. (2017), who showed that lower amounts of linalool over linalyl acetate improved the quality of EO [55]. Moreover, higher amounts of linalyl acetate over 50% strongly decreased the sensory panel score. This fact agreed with the literature, which indicates that the linalyl acetate share in EO should not become excessive by exceeding 46% [20]. 

## 3. Materials and Methods

### 3.1. Plant Material

The study was performed on true lavender flowers cultivated in 2018 in Poland (Kawon-Hurt Nowak Sp.j. Company, Gostyń, Poland). The initial moisture content of the material has been determined for 2.2 kg·kg^−1^. The drying processes were stopped after no further change in weights was observed. The moisture content of samples was determined using a vacuum dryer (SPT-200. ZEAMIL Horyzont, Krakow, Poland). The flowering aerial parts of true lavender were separated from leaves and straws and dried using three methods CD, VMD and CPD-VMFD.

### 3.2. Drying Methods

All drying performances were carried out at the Institute of Agricultural Engineering (Wrocław University of Environmental and Life Sciences, Wrocław, Poland). CD was performed using equipment engineered and constructed therein. True lavender flowers were placed in the round container (d = 100 mm) and dehydrated at 50 °C, 60 °C and 70 °C, all temperatures with an air velocity of 0.5 ms^−1^.

VMD technique was carried on SM 200 dryer (Plazmatronika, Wrocław, Poland). Cylindrical drum (made of glass, 18 cm of diameter × 27 cm of length) was rotated with 6 rev·min^−1^. The vacuum conditions were obtained by applying BL 30P vacuum pump (Tepro, Koszalin, Poland), an MP 211 vacuum gauge (Elvac, Bobolice, Poland), a compensation reservoir of 0.15 m^3^ capacity and a cylindrical tank. Samples were dried with applying three power levels—240, 360 and 480 W, and pressure ranged from 4 up to 6 kPa. The maximum temperature of dried true lavender flowers was measured just after removal from the dryer using an i50 infrared camera (Flir Systems AB, Stockholm, Sweden).

The combinate CPD-VMFD consisted of two steps. First, the material was pre-dried applying CD at 60 °C until the moisture content of flowers approximately 0.44 kg·kg^−1^ was reached. Then the drying was finished by using VMD at 480 W.

### 3.3. Modelling of Drying Kinetics

The drying kinetics of convective drying (CD), vacuum microwave drying (VMD), and combined drying consisting of convective pre-drying followed by vacuum microwave finish-drying (CPD-VMFD) were fitted based on the mass losses of the true lavender samples. The drying kinetics of convective drying (CD) and weight losses were monitored every 2 min for the initial 20 min, and then every 5 min until the end of the drying process.

Vacuum microwave drying (VMD) samples were monitored every 2, 3 and 4 min for 480, 360 and 240 W. Different drying time intervals were applied in order to a similar energy input between subsequent measurements regardless of the microwave power level.

The moisture ratio (MR) of lavender flowers during drying experiments was calculated using the following equation: (1)$$MR=\frac{M_{\left(t\right)}-M_{e}}{M_{o}-M_{e}},$$
where M(t) is the moisture content at time τ, Mo is the initial moisture content, and M_e_ is the equilibrium moisture content (kg water kg^-1^ dw). The values of Me are relatively little compared to those of M_(t)_ or M_o_, the error involved in the simplification is negligible [56,57,58]; thus moisture ratio was calculated as: (2)$$MR=\frac{M_{\left(t\right)}}{M_{0}}.$$

Table Curve 2D Windows v2.03 was used to fit the basic drying models to the measured MR determined according to Equation 2. Several drying models in the literature can be used to describe the kinetics of drying plant materials. For drying model selection, drying curves were fitted to five well-known thin drying models: the modified Page, Henderson–Pabis, logarithmic, Midilli–Kucuk, and Weibull models. The best of fit was determined using two parameters: the values for the coefficient of determination (R2) and root-mean squared error (RMSE). A model fits better if the value of R2 is closer to 1 and the RMSE value is closer to 0 using the following equations:
(3)$$R^{2}=1-[\frac{\sum_{i=1}^{N}{\left(MR_{pre.i}-MR_{exp.i}\right)}^{2}}{\sum_{i=1}^{N}{\left(MR_{per}-MR_{exp.i}\right)}^{2}}],$$
(4)RMSE=[1N∑i=1N(Mexp.i−Mpre.i)]12,
where MR is moisture ratio, (MR) is the mean value of moisture ratio, “pre” and “exp” indicate predicted and experimental values, respectively, while “i” indicates subsequent experimental data, and N is the number of observations. 

Tests conducted in this study proved that the best fitting was obtained for the modified Page model:(5)$$MR=Aexp\left(-k\tau^{n}\right)$$
where A, n and k are constants.

### 3.4. Hydrodistillation of EO

To obtain EOs from true lavender flower samples, Deryng apparatus was applied according to Szumny et al.’s [59] method. Briefly, fresh or dried flowers were placed in 250 mL round bottom flask; then, 150 mL of distilled water was added. The flask was heated up to boiling point and, then, it was kept for 1 h at this temperature. Immediately after reaching the boiling point, 1 mL of cyclohexane with 2 mg of 2-undecanone as internal standard (Sigma-Aldrich, Saint Louis, MO, USA) was added to collect EO. After extraction, a solvent with EO was collected and kept in −18 °C until GC-MS analysis was performed. Hydrodistillations for all samples were run in triplicates.

### 3.5. EO GC-MS Analysis

GC-MS analysis was carried out on Shimadzu GCMS-QP2020 (Shimadzu Company, Kyoto, Japan) equipped with Zebron ZB-5 MSI (30 m × 0.25 mm × 0.25 µm) column (Phenomenex, Torrance, CA, USA). The GC oven temperature was programmed from 50 °C kept for 2 min to 130 °C at a rate of 4.0 °C, then to 270 °C at a rate of 10.0 °C and kept for 5 min. Scanning was performed from 50 to 400 *m/z* in electronic impact (EI) mode at 70 eV. Samples were injected at a 1:100 split ratio, and helium gas was used as the carrier gas at a flow rate of 1.1 mL·min^−1^. Analyses were run in triplicate.

### 3.6. Headspace Solid-Phase Microextraction (HS-SPME)

HS-SPME analysis was applied with 2 cm DVB/CAR/PDMS fibre (Supeclo, Bellefonte, PA, USA). About 0.100 g of fresh flowers or 0.150 g of dried flowers were put into a headspace glass vial and 0.5 mg of 2-undecanone (Sigma-Aldrich, Saint Louis, MO, USA) as an internal standard was added. Then, the vial was placed in a laboratory water bath at 70 °C, and the fibre was exposed above the sample (headspace) for 30 min and followed by analyte desorption in GC injector for 3 min at 220 °C. A GC-MS analysis was performed on Varian CP-3800/Saturn 2000 apparatus (Varian, Walnut Creek, CA, USA) equipped with Zebron ZB-5 MSI (30 m × 0.25 mm × 0.25 µm) column (Phenomenex, Torrance, CA, USA). The GC oven temperature was programmed from 50°C to 130 °C at a rate of 4.0 °C, then to 180 °C at a rate of 10.0 °C, then to 280 °C at a rate of 20.0 °C. Scanning was performed from 35 to 550 *m/z* in electronic impact (EI) mode at 70 eV. Samples were injected at a 1:10 split ratio, and helium gas was used as the carrier gas at a flow rate of 1.0 mL·min^−1^. Analyses were run in triplicate.

### 3.7. Identification and Quantification of Volatile Compounds and EO Constituents

Identification of both volatile compounds and EOs constituents was based upon a comparison of experimentally obtained mass spectra and Kovats retention indices (RI) with those available in NIST WebBook, NIST14 database and literature data [35]. The data was processed using Shimadzu software GCMS Postrun Analysis (Shimadzu Company, Kyoto, Japan) and ACD/Spectrus Processor (Advanced Chemistry Development, Inc., Toronto, ON, Canada). The quantification of identified constituents was performed by calculation based on the amount of added internal standard (2.0 mg of 2-undecanone) and the percentages of particular peaks area. 

### 3.8. Determination of Odour-Active Compounds of True Lavender Flowers

The identification of OACs was performed on Shimadzu GCMS-QP2020 (Shimadzu Company, Kyoto, Japan) with sniffing port, equipped with Restek Rxi-5MS (30 m × 0.25 mm × 0.25 µm) column (Bellefonte, PA, USA). The GC oven temperature was programmed from 50 °C and kept for 2 min, to 130 °C at a rate of 4.0 °C, then to 180 °C at a rate of 10.0 °C, then to 280 °C at a rate of 20.0 °C and kept for 1 min. Scanning was performed from 35 to 550 *m/z* in electronic impact (EI) mode at 70 eV. Samples were injected at a 1:10 split ratio and helium gas was used as the carrier gas at a flow rate of 3.8 mL·min^−1^.

### 3.9. Sensory Evaluation

A group of thirty-nine trained judges was organized at the Orihuela Campus (Escuela Politécnica Superior de Orihuela) of Universidad Miguel Hernández de Elche to evaluate the sensory quality of dried true lavender flowers. The panel was selected and trained following ISO standard 8586-1 [60,61]. Samples were presented coded and in random order in one orientation session. Panellists’ assessments were made in individual booths with controlled illumination (70–90 footcandles) and temperature (23 ± 2 °C). Panellists were asked to rank the samples according to the intensity of sensory attributes (fresh lavender aroma). 

### 3.10. Statistical Analysis

The data from drying kinetics were subjected to the analysis of variance using Tukey’s test (*p* < 0.05) and the data from quantitative EO and volatile constituents were subjected to the analysis of variance using Duncan’s test (*p* < 0.05), all using the STATISTICA 13.3 software for Windows (StatSoft, Krakow, Poland). The data obtained during the sensory panel were analysed using Friedman’s rank-sum analysis (α = 0.05). The means were compared with Tukey’s Honest Significance Difference (HSD) with significance at *p* < 0.05.

## 4. Conclusions

This study on the influence of various drying methods on true lavender (*Lavandula angustifolia* Mill.) flowers cultivated in Poland had revealed that the optimal drying method is highly dependent on the purpose of the dried product. In respect of drying kinetics, it seems that combined CPD-VMFD is the most favourable drying method, nevertheless, in the case of volatile constituents and preferable odour, other methods should be chosen. For highest total EO recovery, the optimal drying method is CD at 50 °C (1.35% of EO in dried true lavender flowers, while in fresh ones it is 5.18%). Also, CD at 50 °C is the optimal drying method for almost all specific major EO constituents. Concerning odour quality, the sensory panel indicated that VMD with power 360 W, as well as combined CPD-VMFD and CD at 50 °C products, possess the strongest aroma compared to fresh true lavender flowers. This result may be caused by specific changes of OACs ratio in the volatile profile of true lavender flowers. HS-SPME analysis had revealed that in those cases of linalool to linalyl acetate, considered as a min OACs of true lavender flowers, the ratio is close to 1:2, which appears to be most favourable distribution.

## Figures and Tables

**Figure 1 molecules-24-02900-f001:**
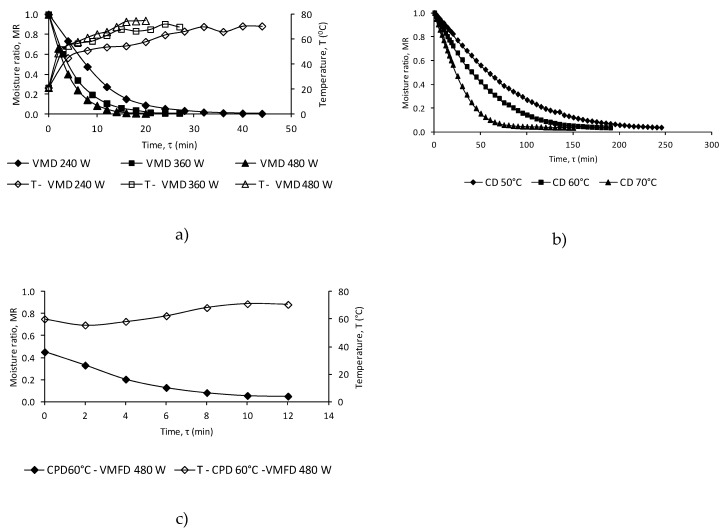
(a) Drying kinetics of true lavender flower samples processed using vacuum microwave drying vacuum-microwave drying (VMD) at powers of 240, 360 and 480 W. (b) Drying kinetics of true lavender flower samples processed using convective drying convective drying (CD) at temperatures of 50, 60 and 70 °C. (c) Drying kinetics of true lavender flower samples processed using vacuum-microwave finish drying (VMFD) at 480 W after convective pre-drying (CPD) at a temperature of 60 °C.

**Figure 2 molecules-24-02900-f002:**
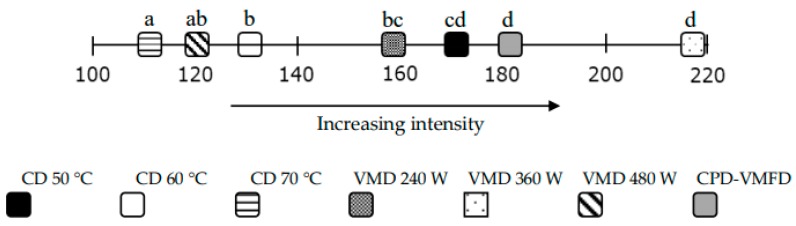
Rank sum scale of lavender aroma intensity. Friedman’s test by Honest Significance Difference (HSD) test (α); different letters after each dried sample indicate a significant difference at *p* < 0.05.

**Table 1 molecules-24-02900-t001:** Final moisture content (M_fwb_), maximum temperature of the sample T, convective drying time (τ), vacuum microwave drying time (τ_1_), and constants A, k and n of the modified Page model describing the drying kinetics.

Drying Conditions	*A*	Constants *k*	*n*	R^2^	RMSE	τ	τ_1_	T(°C)	*M_fwb_ (%)*
CD 50 °C	1	0.0063	1.154	0.9998	0.0049	245	-	50	7.72
CD 60 °C	1	0.0118	1.103	0.9997	0.0062	190	-	60	7.26
CD 70 °C	1	0.0181	1.150	0.9976	0.0193	150	-	70	7.79
VMD 240 W	1	0.0544	1.263	0.9995	0.0071	-	44	64	6.81
VMD 360 W	1	0.1619	1.049	0.9999	0.0033	-	27	73	7.01
VMD 480 W	1	0.1972	1.097	0.9999	0.0025	-	20	75	6.94
CPD 60 °C - VMFD 480 W	0.45	0.2244	1.111	0.9998	0.0072	60	12	71	6.91

**Table 2 molecules-24-02900-t002:** Volatile profile determined by HS, solid-phase microextraction (HS-SPME) of true lavender flowers cultivated in Poland.

Compound	RT [min]	Retention Indices (RI)			Content^4^ [%]	The Match Fitting Score of Obtained Mass Spectra to Mass Spectra Available in Data Base [%]
		RI_exp^1^	RI_lit^2^	RI_lit^3^		
***cis-*** **3-Hexenal**	3.75	801	800	797	tr^5^	91
**cis-** **3-Hexen-1-ol**	4.83	853	857	850	tr	94
**1-Hexanol**	5.09	865	868	863	tr	89
**α-Thujene**	6.59	927	929	924	tr	96
**α-Pinene**	6.79	934	937	932	0.06 ± 0.02	97
**Camphene**	7.22	949	952	946	0.05 ± 0.01	95
**Sabinene**	7.90	973	974	969	tr	94
**1-Octen-3-ol**	8.03	977	979	974	0.13 ± 0.06	97
**Octan-3-one**	8.24	984	986	979	0.50 ± 0.10	97
**β-Myrcene**	8.42	991	991	988	0.23 ± 0.03	97
**3-Octanol**	8.53	995	994	988	0.05 ± 0.01	93
**α-Phellandrene**	8.86	1005	1005	1002	tr	88
**Hexyl acetate**	9.09	1012	1011	1007	0.17 ± 0.05	97
***o*** **-Cymene**	9.39	1021	1022	1022	tr	93
***p*** **-Cymene**	9.48	1024	1025	1020	0.08 ± 0.01	95
**Limonene**	9.61	1028	1030	1024	0.14 ± 0.02	91
**Eucalyptol**	9.70	1030	1032	1026	0.24 ± 0.10	96
***cis*** **-β-Ocimene**	9.90	1037	1037	1032	4.97 ± 0.74	94
***trans*** **-β-Ocimene**	10.20	1046	1048	1044	3.99 ± 0.11	97
**γ-Terpinene**	10.58	1058	1060	1054	0.21 ± 0.08	96
***cis-*** **Sabinene hydrate**	10.88	1067	1070	1065	0.21 ± 0.04	91
***cis*** **-Linalool oxide**	11.06	1072	1074	1067	tr	94
**Terpinolene**	11.62	1089	1088	1086	tr	90
**Linalool**	11.93	1099	1099	1095	8.95 ± 0.48	95
**1-Octen-3-ol acetate**	12.40	1113	1111	1110	1.57 ± 0.20	96
**3-Octanol acetate**	12.78	1123	1124	1120	0.31 ± 0.05	85
**Cosmene**	13.05	1032	1031	-	tr	91
**Camphor**	13.50	1145	1145	1141	0.07 ± 0.02	96
**Hexyl isobutyrate**	13.61	1148	1150	1149	0.07 ± 0.02	95
**Borneol+Lavandulol**	14.34	1170	1170	1165	0.57 ± 0.29	94–95
***1*** **,*trans-3*,*cis-*5-Undecatriene**	14.52	1174	1174	-	0.10 ± 0.01	95
**Terpinen-4-ol**	14.69	1180	1177	1174	3.36 ± 0.29	91
***p*** **-Cymen-8-ol**	14.80	1183	1183	1179	tr	88
**Cryptone**	14.95	1187	1184	1183	0.08 ± 0.02	93
**α-Terpineol**	15.12	1192	1189	1186	0.32 ± 0.04	94
**Myrtenal**	15.28	1197	1193	1195	tr	91
**Octyl acetate**	15.77	1212	1210	1211	tr	94
**Nerol**	16.39	1230	1232	1235	tr	92
**Hexyl 2-methylbutyrate**	16.64	1238	1236	1233	tr	97
**Cumin aldehyde**	16.81	1243	1239	1238	tr	93
**Carvone**	16.94	1247	1242	1239	tr	94
**Linalyl acetate**	17.22	1255	1257	1254	56.94 ± 0.54	96
**Isopulegol acetate**	18.08	1281	1285	1283	0.05 ± 0.01	91
**Bornyl acetate**	18.32	1288	1285	1284	0.30 ± 0.07	92
**Lavandulyl acetate**	18.49	1293	1290	1292	5.17 ± 0.48	96
**Hexyl tiglate**	19.76	1335	1330	1330	tr	96
**α-Longipinene**	20.40	1357	1353	1350	tr	89
***cis*** **-Geranyl acetate**	20.77	1369	1364	1359	0.11 ± 0.02	97
**α-Copaene**	21.13	1382	1376	1374	0.06 ± 0.01	94
***trans-*** **Geranyl acetate**	21.27	1386	1382	1379	0.12 ± 0.02	96
**β-Bourbonene**	21.40	1391	1384	1387	tr	91
**7-*epi*-Sesquithujene**	21.48	1394	1391	1390	0.18 ± 0.03	97
**β-Longipinene**	21.62	1399	1400	1400	tr	89
**Sesquithujene**	21.84	1408	1405	1405	tr	88
**α-Cedrene**	22.05	1419	1411	1410	0.07 ± 0.01	96
**Caryophyllene**	22.19	1426	1419	1417	7.57 ± 1.45	95
***cis*** **-β-Copaene**	22.38	1436	1436	1430	tr	97
***trans*** **-α-Bergamotene**	22.48	1440	1435	1432	0.20 ± 0.06	93
**Unknown**	22.64	1449			tr	-
**Isogermacrene D**	22.72	1453	1448	-	tr	90
**Cadina-3,5-diene**	22.75	1454	1458	-	tr	90
***Allo*** **-aromadendrene**	22.90	1462	1461	1458	3.80 ± 0.76	94
**γ-Muurolene**	23.10	1472	1477	1478	0.05 ± 0.01	92
**Germacrene D**	23.42	1488	1481	1484	0.73 ± 0.20	90
**β-Himachalene**	23.64	1499	1500	1500	tr	91
**Unknown**	23.72	1503			tr	-
**α-Farnesene**	23.83	1510	1508	1505	tr	92
**β-Bisabolene**	23.87	1513	1509	1505	tr	88
**γ-Cadinene**	24.01	1522	1513	1513	0.29 ± 0.11	96
**δ-Cadinene**	24.15	1530	1524	1522	tr	92
**α-Cadinene**	24.39	1546	1538	1537	tr	91
**Caryophyllene oxide**	25.15	1592	1581	1582	tr	94
**Epicubenol**	25.57	1624	1627	1627	tr	95
**τ-Cadinol**	25.85	1645	1640	1635	tr	97

^1^Experimentaly obtained retention indices calculated against *n*-alkanes; ^2^Retention indices according to NIST14 database; ^3^Retention indices according to Adams (2012) [35]; ^4^% calculated from TIC (Total Ion Current) data; ^5^tr < 0.05%.

**Table 3 molecules-24-02900-t003:** Variability of true lavender flower essential oils (EOs) major constituents and total EO affected by various drying methods.

Compound	Drying Method							
	Fresh	CD 50 °C	CD 60 °C	CD 70 °C	CPD-VMFD	VMD 240 W	VMD 360 W	VMD 480 W
	Content [mg 100 g^−1^ dw]^1,2^							
**Octan-3-one**	101.26^a^	20.21^b^	7.90^e^	11.41^de^	15.14^cd^	14.95^cd^	17.85^bc^	17.23^bc^
**Myrcene**	145.56^a^	26.52^b^	12.80^d^	17.07^c^	19.19^c^	11.09^d^	12.16^d^	9.66^d^
***cis*** **-β-Ocimene**	448.38^a^	99.01^b^	57.81^e^	76.89^cd^	80.49^cd^	66.11^de^	88.36^bc^	77.09^cd^
***trans*** **-β-Ocimene**	309.18^a^	69.90^b^	37.09^d^	50.20^c^	50.25^c^	41.45^cd^	43.43^cd^	37.62^d^
**Linalool**	951.51^a^	228.67^bc^	128.01^d^	192.97^c^	249.87^b^	187.99^c^	194.95^c^	193.83^c^
**1-Octen-3-ol acetate**	102.97^a^	21.76^b^	13.87^d^	18.49^c^	15.15^d^	6.62^e^	7.88^e^	6.23^e^
**Borneol+Lavandulol**	133.24^a^	29.69^c^	20.77^d^	21.55^d^	33.77^b^	17.26^e^	9.93^f^	14.99^e^
**Terpinen-4-ol**	367.96^a^	96.42^b^	62.12^de^	83.30^c^	65.32^d^	53.47^e^	73.28^cd^	63.71^de^
**α-Terpineol**	260.85^a^	58.85^b^	34.91^de^	39.44^cd^	44.67^c^	28.24^e^	32.97^de^	25.73^f^
**Linalyl acetate**	983.24^a^	342.22^c^	272.40^d^	377.50^b^	299.99^d^	164.82^f^	203.26^e^	211.33^e^
**Lavandulyl acetate**	358.48^a^	79.85^b^	52.84^d^	79.23^b^	77.80^b^	56.04^cd^	77.91^b^	70.38^bc^
***cis*** **-Geranyl acetate**	144.51^a^	28.29^b^	14.13^d^	18.36^c^	19.41^c^	10.39^de^	11.61^de^	8.53^e^
***trans*** **-Geranyl acetate**	270.15^a^	60.15^b^	32.30^d^	41.68^c^	43.36^c^	24.09^ef^	26.40^e^	19.51^f^
***trans*** **-β-Farnesene**	133.12^a^	31.13^b^	17.47^e^	26.42^c^	21.45^d^	4.44^h^	7.35^g^	9.18^f^
**Total EO [g 100 g^−1^ dw]**	5.18^a^	1.35^b^	0.86^c^	1.16^b^	1.16^b^	0.65^d^	0.90^c^	0.87^c^

^1^Values followed by the same letter within a row are not significantly different (*p* > 0.05, Duncan’s test); ^2^Values based on hydrodistillation using Deryng apparatus.

**Table 4 molecules-24-02900-t004:** Contribution of major volatile compounds in the volatile profile of true lavender flowers affected by various drying methods.

Compound	Drying Method							
	Fresh	CD 50 °C	CD 60 °C	CD 70 °C	CPD-VMFD	VMD 240 W	VMD 360 W	VMD 480 W
	Content [%]^1,2^							
***cis*** **-β-Ocimene**	4.97^a^	1.35^bc^	1.60^b^	1.31^bc^	1.62^b^	1.16^bc^	0.85^c^	1.17^bc^
***trans*** **-β-Ocimene**	3.98^a^	1.10^b^	1.15^b^	0.96^bc^	1.19^b^	0.59^cd^	0.21^d^	0.62^cd^
**Linalool**	8.94^a^	22.17^c^	8.94^a^	14.91^b^	22.03^c^	30.90^e^	24.05^cd^	26.03^d^
**1-Octen-3-ol acetate**	1.57^a^	0.71^c^	1.00^b^	0.85^bc^	0.81^bc^	0.38^d^	0.29^d^	0.29^d^
**Terpinen-4-ol**	3.36^a^	6.44^b^	4.28^a^	3.62^a^	6.10^b^	1.37^c^	3.45^a^	3.45^a^
**Linalyl acetate**	56.94^a^	43.32^d^	58.43^a^	57.48^a^	46.77^b^	44.46^cd^	47.02^b^	46.01^bc^
**Lavandulyl acetate**	5.17^a^	2.92^c^	5.10^a^	4.32^b^	3.34^c^	0.79^e^	1.43^de^	1.64^d^
**Caryophyllene**	7.57^a^	3.88^c^	5.14^bc^	4.46^c^	3.78^c^	6.35^ab^	7.39^a^	6.88^ab^
***Allo*** **-aromadendrene**	3.80^a^	4.91^ab^	5.46^b^	4.18^a^	3.96^a^	1.91^c^	2.27^c^	2.15^c^

^1^Values followed by the same letter within a row are not significantly different (*p* > 0.05, Duncan’s test); ^2^Values based on HS-SPME analyses.

**Table 5 molecules-24-02900-t005:** Odour-active constituents of true lavender flowers and their variability regarding various drying methods.

Compound	Drying Method								Aroma Description
	Fresh	CD 50 °C	CD 60 °C	CD 70 °C	CPD-VMFD	VMD 240 W	VMD 360 W	VMD 480 W	
	Content [%] ^1,2^								
**1-Octen-3-ol**	0.12^ab^	0.26^bc^	0.09^a^	0.13^ab^	0.12^a^	0.27^bc^	0.32^c^	0.15^ab^	Sweet, earthy odor with a strong, herbaceous note reminiscent of lavender–lavandin, rose and hay [53]
**Eucalyptol**	0.24^a^	0.08^ab^	0.08^a^	0.12^a^	0.15^a^	0.58^a^	0.96^b^	0.19^a^	Sweet, sharp vanilla, creamy with spicy clove-like nuances [53]
**Linalool**	8.94^a^	22.17^c^	8.94^a^	14.91^b^	22.03^c^	30.90^e^	24.05^cd^	26.03^d^	Typical pleasant floral odor, free from camphoraceous and terpenic notes [53]
**Borneol+Lavandulol**	0.57^a^	3.10^b^	1.16^b^	1.54^ab^	2.85^b^	1.84^ab^	2.97^b^	2.03^ab^	Floral, waxy, mimosa, herbal [54]
**Terpinen-4-ol**	3.36^a^	6.44^b^	4.28^a^	3.62^ab^	6.10^b^	1.37^c^	3.45^a^	3.45^a^	Cooling, woody, earthy, clove spicy with a citrus undernote [54]
**Nerol**	0.03^a^	0.09^b^	0.05^ab^	0.05^ab^	0.06^ab^	0.08^b^	0.09^b^	0.10^b^	Rosy, slightly citrus, terpy and floral, reminiscent of linalool oxide with aldehydic waxy and fruity nuances [53]
**Linalyl acetate**	56.94^a^	43.32^d^	58.43^a^	57.48^a^	46.77^b^	44.46^cd^	47.02^b^	46.01^bc^	Characteristic bergamot–lavender odor and persistent sweet [53]
**cis-geranyl acetate**	0.11^a^	0.32^d^	0.27^c^	0.23^c^	0.25^c^	0.16^b^	0.16^b^	0.14^ab^	Floral, rosy, sweet, soapy, citrus, grapefruit and fruity with a tropical nuance [54]
***trans*-geranyl acetate**	0.12^a^	0.49^f^	0.39^e^	0.30^cd^	0.37^de^	0.25^bc^	0.23^bc^	0.22^b^	Pleasant, flowery odor reminiscent of rose lavender [53]
**7-epi-Sesquithujene**	0.18^a^	0.11^b^	0.16^a^	0.12^b^	0.10^bc^	0.07^c^	0.07^c^	0.08^c^	Not available

^1^Values followed by the same letter within a row are not significantly different (*p* > 0.05, Duncan’s test); ^2^Values based on HS-SPME analyses.

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
