# Peer review of "Determination of Various Drying Methods’ Impact on Odour Quality of True Lavender (Lavandula angustifolia Mill.) Flowers"

_molecules, 2019, doi:10.3390/molecules24162900_

Round 1

Reviewer 1 Report

As it is stated in the introduction part, the authors continue their almost same previous  investigation on Lavender leaves. Thus, the introduction part is similar in both papers.

There are some inconsistencies regarding  the discussion of results, in particular in 2.2 Volatile profile of true lavender flowers cultivated in Poland. In lines 144-145, a comparison with data reported by Smigielski et al(2018) is presented; however the content of linalool and linalyl acetate determined  in the present study by HS- SPME, is compared with the respective ones  in the lavender Essential Oil [Smigielski et al(2018)], which is a quite different procedure.

The content  of the true lavender flowers Eos major constituents in table 3 should be presented as percentages, in order  to be comparable with those in Table 2 and Table 4, presented in different units (HS-SPME analyses)

In the present study the essential oil was   isolated by hydrodistillation, by using the Deryng apparatus. However, the lavender essential oil is generally obtained by steam distillation. Consequently the two methods, influence differently the content of the main compounds in the obtained essential oil. The authors should take into consideration the  different effect of hydrodistillation and steam distillation on the essential oil.

Moreover, in most of the European  countries Pharmacopoeas, the essential oil determination by hydrodistillation, is recommended to be carried out by the Clevenger type apparatus.

Taking into consideration that the majority of the lavender production goes for  essential oil, which is  obtained by steam distillation of fresh flowers, the objectives of this study, i.e the effect of drying  methods  (which is usually demanding high energy consumption), would be more matched for a research on  the preservation of the raw material and its quality characteristics

Author Response

Thank you for your comments and advices. We consider your suggestions as highly improving the quality of our work. Please find our responses and comments below, as well as in word file. We hope that you will be satisfied with our improvements.

“There are some inconsistencies regarding the discussion of results, in particular in 2.2 Volatile profile of true lavender flowers cultivated in Poland. In lines 144-145, a comparison with data reported by Smigielski et al(2018) is presented; however the content of linalool and linalyl acetate determined  in the present study by HS- SPME, is compared with the respective ones  in the lavender Essential Oil [Smigielski et al(2018)], which is a quite different procedure.”

Dear Reviewer, thank you for this observation. We totally agree that this values should not be compared. We are sorry for this oversight. To correct that, we had moved the part regarding the amount of linalyl acetate and linalool and the discussion with Śmiegielski et al. (2018) results to 2.3 (lines 180-184), where we focus on the real EO amounts, and the share of particular constituents in it. We also had verified the percentages of linalyl acetate and linalool in relation to total EO. We hope that it will satisfy you.

“The content of the true lavender flowers Eos major constituents in table 3 should be presented as percentages, in order  to be comparable with those in Table 2 and Table 4, presented in different units (HS-SPME analyses).”

Dear Reviewer, thank you for this suggestion, however in our opinion changing the form of presentation of this result fill overcome the data related to this results. Our purpose was to present the clear information about real content of particular major constituents, that raise the most interest. Presenting this for percentages, whether in relation to total EO content or dry weight matter, will not provide the readers clear information. The differences apparent now, will be flattened. We hope that you will approve our point of view.

“In the present study the essential oil was isolated by hydrodistillation, by using the Deryng apparatus. However, the lavender essential oil is generally obtained by steam distillation. Consequently the two methods, influence differently the content of the main compounds in the obtained essential oil. The authors should take into consideration the  different effect of hydrodistillation and steam distillation on the essential oil.”

Dear Reviewer, thank you for this accurate insight. We had provide in manuscript (lines 177-180) the justification of this procedure supported by some references.

Moreover, in most of the European countries Pharmacopoeas, the essential oil determination by hydrodistillation, is recommended to be carried out by the Clevenger type apparatus.

We fully agree, that in European Pharmacopoeia 9 the Clevenger-type apparatus is recommended for isolation and  quantification of essential oil from plant material. However, Deryng system was presented in Pharmacopoeias in Central Europe countries. We must underline at this point, that Clevenger-type apparatus was invented in 1928 by Joseph Franklin Clevenger and it is used up to now. From the other hand, some inconveniences of this invention (e.g. vent bellow cooler and huge fragility) was removed in Deryng apparatus (Polish patent by Jakub Deryng, published further in  paper Deryng, J. (1951). Nowy aparat do oznaczanie olejków w materiale roślinnym (A new apparatus for determining oils in plant material). Acta Pol. Pharm, 8, 121-136 (in polish). European Pharmacopoeia  9 ordered unification of equipment according to rules in so called west European countries. According to our  experience hydrodistillation on Deryng apparatus gives more accurate results comparing to Clevenger. Moreover above 25 papers by Antoni Szumny or Angel Antonio Carbonell-Barrachina group has been published with using Deryng-type apparatus for hydrodistillation. 

Reviewer 2 Report

The manuscript submitted for publication in Molecules refers to the assay of the impact of the drying method on the odor quality of the true Lavender flowers. Convective drying, vacuum microwave drying and combined convective/vacuum microwave drying methods were compared. Head space - SPME/GC/MS was used for assaying the volatile compounds profile in the essential oil extracted from the vegetal material. Additionally, a sensory panel was used to asses the samples odor quality, through GC-O.

The submitted manuscript is a mirror image of a previous published material of the authors in Molecules, dealing with the impact of the drying methods on the odor quality of the Lavender leaves (see reference 25). The olfactory analysis was added to the new submitted manuscript. Strictly, from the analytical point of view, the submitted manuscript does not add novelty with respect to the previous published results. The conclusions relating to the drying method of choice, although correct, are not surprising at all. However, the findings (including the olfactory analysis) may be useful for the industry relating to the production of essential oils derived from Lavender.

I suggest that in Table 2 authors should add a new column relating to the identification of compounds by means of the comparison between the experimental mass spectra resulting from the analysis and mass spectra existing in the spectral library (the match fitting score). Additionally, a verification of the English language throughout the text is needed.

To conclude, the manuscript may be accepted for publication with a moderate revision, assuming that the novelty and intrinsic original characteristics of the material are not evident.

Author Response

Thank you for your comments and advices. We consider your suggestions as highly improving the quality of our work. Please find our responses and comments below, as well as in word file. We hope that you will be satisfied with our improvements.

I suggest that in Table 2 authors should add a new column relating to the identification of compounds by means of the comparison between the experimental mass spectra resulting from the analysis and mass spectra existing in the spectral library (the match fitting score).

Dear Reviewer, thank you for this comment. We had provided in Table 2 the additional column, with requested data.

Reviewer 3 Report

 This paper evaluated the effects of different drying methods to the content of volatile odor-active constituents in true lavender flowers. The experiments were carefully conducted and results explained. However, many misspellings and grammatical errors occurred throughout, which strongly impacted the quality of the paper. Please get professional help on the punctuation, preposition, articles, etc.

Here are some concerns:

1.       How were the volatile constituents quantified by GC-MS? Section 3.7 only described the identification method. How about the reported values in the literature that were used for comparison in the Results? Did they use 2-undecanone as internal standard as well? Will different quantification methods impact the results comparison?

2.       The acronym should be defined after the first appearance in the text with parentheses. Once defined, only the acronym should be used. For example, VMD and CD were defined in the legend of Figure 1, page 3, where these two words have been defined in page 2. EO was defined in line 45 but it was spelled out in line 57.

3.       The measurement in Figure 1 represent how many replicates?

4.       What are the superscripts a to f in Table 3? Are they the aroma intensity scales showing in Figure 2? Please explain.

5.       Please remake the figure in the SI file “List of all constituents of true lavender essential oils”. Please add compound names on top of the GC peaks, and re-scale the figure width to one page.  

6.       Please remove the extra page in the SI file “Sensory panel ranks”.

7.       Since each SI file is small, can the authors combine them into one SI file with table of content, table titles and figure legends?

Some minor suggestions:

1.       Please define GC-MS-O in the abstract

2.       What is DVB/CAR/PDMS (Page 6 line 155, page 14 line 325)

3.       There is no “(c)” in the legend of Figure 1.

4.       The number of Conclusions should be 4, not 5.

Author Response

Thank you for your comments and advices. We consider your suggestions as highly improving the quality of our work. Please find our responses and comments below, as well as in word file. We hope that you will be satisfied with our improvements.

How were the volatile constituents quantified by GC-MS? Section 3.7 only described the identification method. How about the reported values in the literature that were used for comparison in the Results? Did they use 2-undecanone as internal standard as well? Will different quantification methods impact the results comparison?

Dear Reviewer, thank you for this comment. We had added the description of the quantification method in lines 347-349. In the case of other studies, the procedure are diversified. It is very common to only present the share [%] of particular constituents in total EO. Nevertheless, procedure we applied is also used in some studies that we referred to, regarding different plants and EO analyses. We had chosen 2-undecanone thus our experience and peaks intensity high repeatability. The differences in quantification efficiency of course may occur, although also different gc-ms apparatus, its calibration, applied ionization method may case those. That’s why we try to refer to paper published in renowned journals.

The acronym should be defined after the first appearance in the text with parentheses. Once defined, only the acronym should be used. For example, VMD and CD were defined in the legend of Figure 1, page 3, where these two words have been defined in page 2. EO was defined in line 45 but it was spelled out in line 57.

Dear Reviewer, thank you for this insight. We verified the usage of acronyms in the manuscript.

The measurement in Figure 1 represent how many replicates?

The measurement in Figure 1 was done as single measurement, due to long time of drying process, which causes degradation of analysed volatile compounds in harvested plants, which were not used at the moment. Also equipment shortage did not allowed as to perform more replications. That is way, we decided to run one measurement to perform all dryings on same quality raw material.

What are the superscripts a to f in Table 3? Are they the aroma intensity scales showing in Figure 2? Please explain.

Dear Reviewer, the superscripts in all tables or figures corresponds to statistical analysis. They indicate that in particular table or figure numbers followed be the same letter are not statistically different. They should not be compared between tables and figures.

“Please remake the figure in the SI file “List of all constituents of true lavender essential oils”. Please add compound names on top of the GC peaks, and re-scale the figure width to one page; 6. Please remove the extra page in the SI file “Sensory panel ranks.”; 7. Since each SI file is small, can the authors combine them into one SI file with table of content, table titles and figure legends?”

Dear Reviewer, thank you for this suggestions. We had prepared just one file with supplementary materials, according to your comment. Also, we had resized the chromatogram, however to make it more clear, we had singed just the major constituents peaks. We hope that it will satisfy you.

Some minor suggestions:

Please define GC-MS-O in the abstract

What is DVB/CAR/PDMS (Page 6 line 155, page 14 line 325)

There is no “(c)” in the legend of Figure 1.

The number of Conclusions should be 4, not 5.

Dear Reviewer, thank you for all helpful minor suggestions. All of them were addressed.

Round 2

Reviewer 1 Report

thanks for addressing my comments

Reviewer 3 Report

Thanks for addressing my comments.